# Bionic Tracking: Using Eye Tracking to Track Biological Cells in Virtual Reality

**Abstract.** We present Bionic Tracking, a novel method for solving biological cell tracking problems with eye tracking in virtual reality using commodity hardware. Using gaze data, and especially smooth pursuit eye movements, we are able to track cells in time series of 3D volumetric datasets. The problem of tracking cells is ubiquitous in developmental biology, where large volumetric microscopy datasets are acquired on a daily basis, often comprising hundreds or thousands of time points that span hours or days. The image data, however, is only a means to an end, and scientists are often interested in the reconstruction of cell trajectories and cell lineage trees. Reliably tracking cells in crowded three-dimensional space over many timepoints remains an open problem, and many current approaches rely on tedious manual annotation and curation. In our Bionic Tracking approach, we substitute the usual 2D point-and-click annotation to track cells with eye tracking in a virtual reality headset, where users simply have to follow a cell with their eyes in 3D space in order to track it. We detail the interaction design of our approach and explain the graph-based algorithm used to connect different time points, also taking occlusion and user distraction into account. We demonstrate our cell tracking method using the example of two different biological datasets. Finally, we report on a user study with seven cell tracking experts, demonstrating the benefits of our approach over manual point-and-click tracking.

## 1 Introduction

In cell and developmental biology, the image data generated via fluorescence microscopy is often only a means to an end: Many tasks require exact information about the positions of cells during development, or even their entire history, the so-called cell lineage tree. Both the creation of such a tree using cell tracking, and tracking of single cells, are difficult and cannot always be done in a fully automatic manner. Therefore, such lineage trees are created in a tedious manual process using a point-and-click 2D interface. Even if cells can be tracked (semi)automatically, faulty tracks have to be repaired manually. Again, this is a very tedious task, as the users have to go through each timepoint and 2D section in order to connect cells in 3D+time, with a 2D point-and-click interface. Manually tracking one single cell through 101 timepoints with this manual process takes 5 to 30 minutes, depending on complexity of the dataset. Tracking an entire developmental dataset with many 3D images can take months of manual curation effort.

The 3D images the lineage trees are usually created based on fluorescence microscopy images. Such fluorescence images do not have well-defined intensity scales, and intensities might vary strongly even within single cells. Cells also move around, divide, change

their shape—sometimes drastically—or might die. Cells might also not appear alone, and may move through densely-populated tissue, making it difficult to tell one cell apart from another. These three issues are the main reasons that make the task of tracking cells so difficult. Further complicating the situation, recent advances in fluorescence microscopy, such as the advent and widespread use of lightsheet microscopy [9], have led to a large increase in size of the images, with datasets growing from about a gigabyte to several terabytes for long-term timelapse images [25].

In this work, we reduce the effort needed to track cells through time series of 3D images by introducing *Bionic Tracking*, a method that uses smooth pursuit eye movements as detected by eye trackers inside a virtual reality head-mounted display (HMD) to render cell tracking and track curation tasks easier, faster, and more ergonomic. Instead of following a cell by point-and-click, users have to simply look at a cell in Virtual Reality (VR) in order to track it. The main contributions we present here are:

– A setup for interactively tracking cells by simply following the cell in a 3D volume rendering with the eyes, using a virtual reality headset equipped with eye trackers,
– an iterative, graph-based algorithm to connect gaze samples over time with cells in volumetric datasets, addressing both the problems of occlusion and user distraction, and
– a user study evaluating the setup and the algorithm with seven cell tracking experts

## 2   Related Work

The main problem we address in this paper is the manual curation or tracking step, which is necessary for both validation and for handling cases where automatic tracking produces incorrect or no results. In this section, we give a brief overview of (semi-)automatic tracking algorithms, then continue with relevant work from the VR, visualization, and eye tracking communities.

Historically, software for solving tracking problems was developed for a specific model organism, such as for the roundworm *Caenorhabditis elegans*, the fruitfly *Drosophila melanogaster*, or the zebrafish *Danio rerio* — all highly studied animals in biology — and relied on stereotypical developmental dynamics within an organism in order to succeed in tracking cells. This approach however either fails entirely or produces unreliable results for other organisms, or for organisms whose development is not as stereotyped. For that reason, (semi-)automated approaches have been developed that are independent of the model organism and can track large amounts of cells, but often require manual tracking of at least a subset of the cells in a dataset. Examples of such frameworks are:

– *TGMM*, Tracking by Gaussian Mixture Models [2, 1], is an offline tracking solution that works by generating oversegmented supervoxels from the original image data, then fit all cell nuclei with a Gaussian Mixture Model and evolve that through time, and finally use the temporal context of a cell track to create the lineage tree.
– *TrackMate* [31] is a plugin for Fiji [26] that provides automatic, semi-automatic, and manual tracking of single particles in image datasets. TrackMate can be extended with custom spot detection and tracking algorithms.

- *MaMuT*, the Massive MultiView Tracker [36], is another plugin for Fiji that allows the user to manually track cells in large datasets, often originating from multi-view lightsheet microscopes. MaMuT's viewer is based on BigDataViewer [22] and is able to handle terabytes of data.

All automated approaches have in common that they need manual curation as a final step, as they all make assumptions about cell shapes, modelling them, e.g., as blobs of Gaussian shape, as in the case of TGMM.

Manual tracking and curation is usually done with mouse-and-keyboard interaction to select a cell and create a track, often while just viewing a single slice of a 3D time point of the dataset. In Bionic Tracking, we replace this interaction by leveraging the user's gaze in a virtual reality headset, while the user can move freely around or in the dataset. Gaze in general has been used in human-computer interaction for various interactions: It has been used as an additional input modality in conjunction with touch interaction [29] or pedaling [14], and for building user interfaces, e.g., for text entry [19].

The particular kind of eye movements we exploit for Bionic Tracking—*smooth pursuits*, where the eyes follow a stimulus in a smooth, continuous manner—is not yet explored exhaustively for interacting with 3D or VR content. Applications can be found mainly in 2D interfaces, such as in [15], where the authors use deviations from smoothness in smooth pursuits to evaluate cognitive load; or in [34], where smooth pursuits are used for item selection in 2D user interfaces. For smooth pursuits in VR, we are only aware of two works, [24] and [13]: In the first, the authors introduce *Radial Pursuit*, a technique where the user can select an object in a 3D scene by tracking it with her eyes, and it will become more "lensed-out" the longer she focuses on a particular object. In the latter, the authors explore target selection using smooth pursuits, perform a user study, and make design recommendations for smooth pursuit-based VR interfaces.

All aforementioned works are only concerned with navigation or selection tasks on structured, geometric data. In Bionic Tracking however, we use smooth pursuits to track cells in unstructured, volumetric data that cannot simply be queried for the objects contained or their positions.

In the context of biomedical image analysis, VR has been applied successfully, e.g., for virtual colonoscopy [21] and for tracing of neurons in connectome data [33]. In the latter, the authors show the neurons in VR in order to let the user trace them with a handheld controller. The authors state that this technique resulted in faster and better-quality annotations. Tracking cells using handheld VR controllers is an alternative to gaze, but could place higher physical strain on the user.

## 3   The Bionic Tracking Approach

For Bionic Tracking, we exploit smooth pursuit eye movements. Smooth pursuits are the only smooth movements performed by our eyes. The occur when following a stimulus, and cannot be triggered without one [6]. Instead of using a regular 2D screen, we perform the cell tracking process in VR, since VR gives the user improved navigation and situational awareness compared to 2D when exploring a complex 3D/4D dataset [28].

In addition, the HMD tracking data can be used to impose constraints on the data acquired from the eye trackers. In order to remove outliers from gaze data one can

calculate the quaternion distance between eyeball rotation and head rotation, which is physiologically limited: a 90-degree angle between eye direction and head direction is not plausible, and head movement follows eye movement via the vestibo-ocular reflex.

As a system consisting of both a VR HMD and an integrated eye tracking solution might be perceived as too complex, we start by explaining why we think that only using one of the technologies would not solve the problem:

– *Without eye tracking*, the head orientation from the HMD could still be used as a cursor. However, following small and smooth movements with the head is not something humans are used to doing. The eyes always lead the way, and the head follows via the vestibulo-ocular reflex.
– *Without virtual reality*, the effective space in which the user can use to follow cells around becomes restricted to the rather small part of the visual field a regular screen occupies. The user furthermore loses the ability to move around freely without an additional input modality, e.g. to avoid obstacles (in our case, those might be cells not tracked at the moment). As an alternative to HMDs, a system using large screens or projectors, such as Powerwalls or CAVEs, could be used, but increases the technical complexity.

### 3.1   Hardware selection

We have chosen the HTC Vive as HMD, as it is comfortable to wear, provides good resolution, and an excellent tracking system for room-scale VR experiences. Furthermore, it is usable with the SteamVR/OpenVR API. For eye tracking, we have chosen the *Pupil* eye trackers produced by Pupil Labs [11], as they provide both an open-source software and competitively-priced hardware that is simple to integrate physically into off-the-shelf HMDs. The software is available as LGPL-licensed open-source code and can be extended with custom plugins.

In addition to being open-source, the *Pupil* software makes the measured gaze data and image frames available to external applications via a simple ZeroMQ- and MessagePack-based protocol[1]—in contrast to closed-source proprietary libraries required by other products—which enables using the eye tracking data in a local application or even over the network.

Alternative solutions, like the HTC Vive Pro Eye, or an HTC Vive with integrated Tobii eye tracker were either not available at the time this project started, or were much more expensive.

### 3.2   Software framework

We have developed Bionic Tracking using the visualization framework *scenery* [7], as it supports rendering of mesh data simultaneously with multi-timepoint volumetric data that contains the cells or nuclei to be tracked. Crucially for Bionic Tracking, scenery supports rendering to all SteamVR/OpenVR-supported VR HMDs and supports the

---

[1] See https://docs.pupil-labs.com/developer/core/network-api/ for details on interacting with Pupil over the network.

Pupil eye trackers. In addition, scenery runs on the Java VM and is interoperable with the image analysis toolkit Fiji, just as the existing tracking tools *TrackMate* and *MaMuT* (see Section 2).

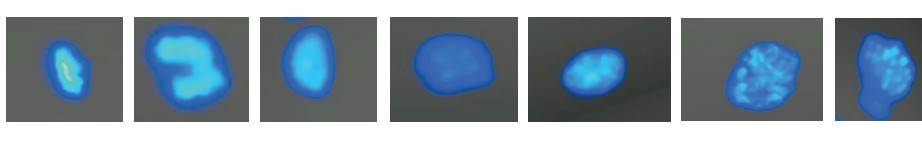

**Fig. 1:** Some example nucleus shapes encountered in our *Platynereis* test dataset.

### 3.3   Rendering

We use simple, alpha blending-based volume rendering for displaying the data in the VR headset using scenery's Vulkan backend. While more advanced algorithms for volume rendering exist which provide a higher visual quality (e.g. Metropolis Light Transport [16]), achieving a high and ideally consistent framerate is important for VR applications, which led us to choose alpha blending. For the data used in this work, we have only used in-core rendering, while the framework also supports out-of-core volume rendering for even larger datasets. To the user, we not only display the volume on its own, but a gray, unobtrusive box for spatial anchoring around the volume (see the supplementary video for an impression of how this looks).

## 4   Tracking Cells with Bionic Tracking

### 4.1   Preparation

After putting on the VR HMD, making sure the eye tracker's cameras can see the user's eyes and launching the application, the calibration routine needs to be run first in order to establish a mapping between the user's gaze and world space positions in the VR scene. For calibration, we show the user a total of 18 white spheres, with 5 of them layered on three circles 1 m apart (distances in the VR scene are the same as in the physical world). The radius of the circles increases with each layer to achieve a good coverage of the field of view. In addition to the spheres on the circles, we show three spheres in the center of the circles to also cover the area in the center of the field of view. During the calibration routine, the user has to look at these spheres as they are shown in the HMD. Since the calibration targets follow the head movements of the user, the user does not need to stay still. At the end of the calibration, the user will be notified of success or failure, and can repeat the calibration process if necessary. Calibration typically needs to be run only once per session, and can then be used to track as many cells as the user likes. Exceptions are if there is significant slippage or if the HMD is removed during the session. Our calibration routine is mostly similar to the one used in *Pupil's* HMDeyes Unity example project[2].

Movement in VR can be performed either physically, or via buttons on the handheld controllers, which additionally allow control of the following functions (handedness can be swapped, default bindings shown in Supp. Fig. 1):

---

[2] See https://github.com/pupil-software/hmd-eyes for details.

- move the dataset by holding the left-hand trigger and moving the controller,
- use the directional pad on the left-hand controller to move the observer (forward, backward, left, or right – with respect to the direction the user is looking to),
- start and stop tracking by pressing the right-hand side trigger,
- deleting the most recently created track by pressing the right-side button, and confirming within three seconds with another press of the same button,
- play and pause the dataset over time by pressing the right-hand menu button,
- play the dataset faster or slower in time by pressing the right-hand directional pad up or down, and
- stepping through the timepoints of the dataset one by one, forward or backward, by pressing the right-hand directional pad left or right.

When the dataset is not playing, the user can also use the directional pad on the right-hand controller to scale the dataset. The initial setting for the scale of the dataset is to make it appear about 2m big.

## 4.2 Tracking Process

After calibration, the user can position herself freely in space. To track a cell, the user performs the following steps:

1. Find the timepoint and cell with which the track should start, adjust playback speed between one and 20 volumes/second, and start to look at the cell or object of interest,
2. start playback of the multi-timepoint dataset, while continuing to follow the cell by looking at it, and maybe moving physically to follow the cell around occlusions,
3. end or pause the track at the final timepoint. Tracking will stop automatically when playback as reached the end of the dataset, and the dataset will play again from the beginning.

In order to minimize user strain in smooth pursuit-based VR interactions, the authors of [13] have provided design guidelines: They suggest large trajectory sizes, clear instructions what the user has to look at, and relatively short selection times. While physical cell size cannot be influenced, the controls available to the user enable free positioning and zooming. The selection time, here the tracking time, of course depends on the individual cell to be tracked, but as the tracking can be paused, and the playback speed adjusted, the user is free to choose both a comfortable length and speed.

During the tracking procedure, we collect the following data for each timepoint:

- the entry and exit points of the gaze ray through the volume in normalised volume-local coordinates, i.e., as a vector $\in [0.0, 1.0]^3$,
- the confidence rating – calculated by the *Pupil* software – of the gaze ray,
- the user's head orientation and position,
- the timepoint of the volume, and

– a list of sampling points with uniform spacing along the gaze ray through the volume and the actual sample values on these points calculated by trilinear interpolation from the volume image data.

We call a single gaze ray including the above metadata a *spine*. The set of all spines for a single track over time we call a *hedgehog* – due to its appearance, see Supp. Fig. 2. By collecting the spines through the volume, we are effectively able to transform each 3-dimensional cell localization problem into a 1-dimensional one along a single ray through the volume and create a cell track. This analysis procedure is explained in detail in the next section.

## 5   Analysis of the Tracking Data

In previous applications using smooth pursuits (such as in [34, 24]), the tracked objects were geometric and not volumetric in nature, and therefore well-defined in 2D or 3D space with their extents and shape fully known. In our analysis in contrast, we use the indirect information about the objects contained in spines and hedgehogs to find the tracked object in unstructured volumetric data and follow it.

After a full hedgehog has been collected to create a new cell track, all further analysis is done solely on the data contained in this hedgehog. To illustrate the analysis, it is useful to visualize a hedgehog in two dimensions by laying out all spines in a 2D plane next to each other (see Figure 2). In this plane, time advances along the X axis and depth through the volume along a given spine is on the Y axis. Note that each line parallel to the Y axis represents one spine and therefore one gaze sample, of which we collect up to 60 per second. In Figure 2, this led to 1614 spines with 16 spines per image timepoint on average collected within 30 seconds. In the figure, we have highlighted the local intensity maximum along each spine in red. The track of the cell the user was following is then mostly visible.

### 5.1   Graph-based temporal tracking

Movements of the user and temporary occlusion by other cells or objects render it challenging to reliably extract a space-time trajectory from the information contained in the hedgehog. In order to reliably link cell detections across timepoints, we use an incremental graph-based approach based on all spines that have local maxima in their sample values. A plot of an exemplary spine through a volume is shown in Supp. Fig. 3. In the figure, the distance from the observer in voxels along the spine is shown on the X axis, while the Y axis shows the intensity value of the volume data at that point along the spine. To initialize the algorithm, we assume that when starting a track the user looks at an unoccluded cell that is visible as the nearest local maximum along the spine. In Supp. Fig. 3 that would be the leftmost local maximum.

For each timepoint, we have collected a variable number of spines, whose count varies between 0 and 120; zero spines might be obtained in case that the user closes her eyes, or that no detection was possible for other reasons, and 120 Hz is the maximum frame rate of the eye trackers used.

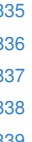
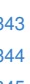
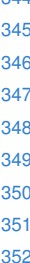

**Fig. 2:** A hedgehog visualized in 2D, with nearest local maxima marked in red. Each vertical line is one spine of the hedgehog with the observer sitting at the bottom. On the X axis, time runs from left to right, and is counted in gaze samples taken. After every 500 spines, a dotted white line is shown at 500, 1000, and 1500 spines recorded. The gray line shortly before 500 spines is the line whose profile is shown in Supp. Fig. 3. The discontinuities in the local maxima A and B have different origins: For A, the user seems to have moved further away, resulting in a gap, while for B, another cell appeared closely behind the tracked one and might have mislead the user, leaving it for the algorithm to filter out. See text for details.

In order to correctly track a cell across spines over time, and after the initial seed point on the first spine has been determined, we step through the spines in the hedgehog one by one, performing the following operations, as illustrated in Figure 3:

1. advance to the next spine in the hedgehog,
2. find the indices of all local maxima along the spine, ordered by world-space distance to the selected point from the previous spine,
3. connect the selected point from the previous spine with the closest (in world-space distance) local maximum in the current spine,
4. calculate the world-space position of the new selected point, and
5. add the selected point to the set of points for the current track.

In addition to connecting discontinuities in the local maxima detected (discontinuity A in Figure 2) world-space distance weighting also excludes cases where another cell is briefly moving close to the user and the actually tracked cell (discontinuity B in Figure 2). The process of connecting a local maximum to the nearest one at a later time is a variant of *dynamic fringe-saving A\** search on a grid [30] with all rays extended to the maximum length in the entire hedgehog along the X axis, and time increasing along the Y axis.

This strategy constructs a cell track from the spines of each hedgehog. The calculation of the final track typically takes less than a second and is visualised right away, such that the user can quickly decide whether to keep it, or discard it.

## 5.2 Handling Distraction and Occlusions

In some cases, however, world-space distance weighting is not enough, and a kind of Midas touch problem [10] remains: When the user briefly looks somewhere else than at the cell of interest, and another local maximum is detected there, that local maximum

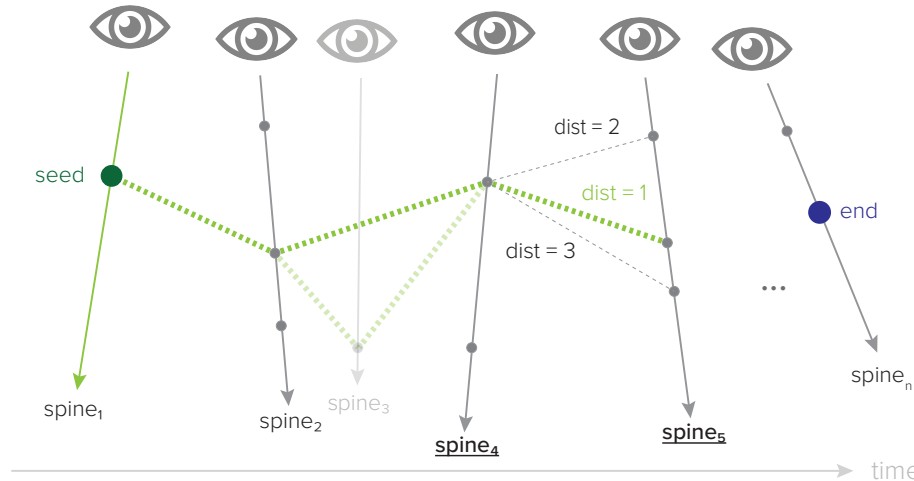

**Fig. 3:** A graphical illustration of the incremental graph-search algorithm used to extract tracks from a hedgehog. Time runs along the X axis. $spine_1$ contains the initial seed point where to start tracking. The algorithm is currently at $spine_4$, determining how to proceed to $spine_5$. In this case, the middle track with dist = 1 wins, as it is the shortest world-space distance away from the current point. The algorithm will continue the path search until it has reached the last spine, $spine_n$. In this manner, the algorithm closes the gaps around the sample numbers 700 and 1200 in Figure 2, and leaves out the detected cells further along the individual rays. $spine_3$ is connected initially, but removed in the final statistical pruning step. It is therefore grayed out. See text for details.

may indeed have the smallest world-space distance and win. This would introduce a wrong link in the track. Usually, the Midas touch problem is avoided by resorting to multimodal input (see, e.g., [29, 20]). Here, we aim to avoid the Midas touch problem without burdening the user with additional modalities of control. We instead use statistics: for each vertex distance $d$, we calculate the z-score $Z(d) = (d - \mu_{dist})/\sigma_{dist}$, where $\mu_{dist}$ is the mean distance in the entire hedgehog and $\sigma_{dist}$ is the standard deviation of all distances in the entire hedgehog. We then prune all graph vertices with a z-score higher than 2.0. This corresponds to distances larger than double the standard deviation of all distances the hedgehog. Pruning and graph calculations are repeated iteratively until no vertices with a z-score higher than 2.0 remain, effectively filtering out discontinuities like B in Figure 2.

## 6   Proof of concept

We demonstrate the applicability of the method with two different datasets:

- A developmental 101-timepoint dataset of a *Platynereis dumerilii* embryo, an ocean-dwelling ringworm, acquired using a custom-built OpenSPIM [23] lightsheet microscope, with cell nuclei tagged with the fluorescent GFP protein (16bit stacks, 700x660x113 pixel, 100MB/timepoint, 9.8 GByte total size),

– A 12-timepoint dataset of *MDA231* human breast cancer cells, embedded in a collagen matrix and infected with viruses tagged with the fluorescent GFP protein, acquired using a commercial Olympus FluoView F1000 confocal microscope (dataset from the Cell Tracking Challenge [32], 16 bit TIFF stacks, 512x512x30 pixels, 15MB/timepoint, 98 MByte total size).

The *Platynereis* dataset was chosen because it poses a current research challenge, with all tested semiautomatic algorithms failing on this dataset, due to the diverse nuclei shapes and cell movements. Examples of shapes encountered in the dataset are shown in Figure 1. The MDA231 dataset in turn was chosen because it had the worst success scores for automatic tracking methods on the `https://celltrackingchallenge.net` website due to the diversity of cell shapes and jerky movements in the dataset.

For the *Platynereis* dataset, we were able to quickly obtain high-quality cell tracks using our prototype system. A visualization of one such cell track is shown in Supplementary Figure 4. In the companion video, we show both the gaze tracking process to create the track and a visualization showing all spines used to generate the track.

For the MDA231 dataset, we are able to obtain tracks for six moving cells in the dataset in about 10 minutes. A visualization of these tracks is shown in Supp. Fig. 5; see the companion video for a part of the tracking process. This example also demonstrates that the Bionic Tracking technique is useful even on nearly "flat" microscopy images in VR, as the dataset only has 30 Z slices, compared to a resolution of 512x512 in X and Y.

All datasets are rendered at their full resolution, with a typical framerate of 60-90fps.

## 7    Evaluation

We evaluated Bionic tracking by first performing a user study to gain insight into user acceptance and feasibility. We then compared tracks created with Bionic Tracking to the manually annotated ground truth. Together, these evaluations serve as an initial characterization of the usability and performance of Bionic Tracking.

### 7.1    User Study

We recruited seven cell tracking experts who were either proficient with manual cell tracking tasks in biology, proficient in using or developing automated tracking algorithms, or both (median age 36, s.d. 7.23, 1 female, 6 male) to take part in the study. The users were given the task to track arbitrary cells in the *Platynereis* dataset already used in Section 6. One of the users was already familiar with this particular dataset. The study was conducted on a Dell Precision Tower 7910 workstation (Intel Xeon E5-2630v3 CPU, 8 cores, 64 GB RAM, GeForce GTX 1080Ti GPU) running Windows 10, build 1909.

Before starting to use the software, all users were informed of the goals and potential risks (e.g., simulator sickness) of the study. With a questionnaire, they were asked for presence of any visual or motor impairments (apart from needing to wear glasses or contact lenses, none were reported), about previous VR experience and physical wellbeing. After using the software, users were again asked about their physical wellbeing, and had to judge their experience using the NASA Task Load Index (TLX, [8]) and

Simulator Sickness Questionnaire (SSQ, [12]). In addition, they were asked both quali-
tative and quantative questions about the software based on both the User Experience
Questionnaire [17] and the System Usability Scale [3]. We concluded the study for each
participant with a short interview where users were asked to state areas of improvement,
and what they liked about the software. The full questionnaire used in the study is
available in the supplementary materials.

After filling the pre-study part of the questionnaire, users were given a brief introduc-
tion to the controls in the software. After ensuring a good fit of the HMD on the user's
head, the interpupillary distance (IPD) of the HMD was adjusted to the user's eyes, as
were the ROIs of the eye tracking cameras. The users then ran the calibration routine
on their own. Then, they were able to take time to freely explore the dataset in space
and time. If the calibration was found to not be sufficiently accurate, we re-adjusted
HMD fit and camera ROIs, and ran the calibration routine again. Finally, all users were
tasked with tracking the cells in the *Platynereis* dataset. Users were then able to create
cell tracks freely, creating up to 32 cell tracks in 10 to 29 minutes.

All participants in the study had no or very limited experience with using VR
interfaces (5-point scale, 0 means no experience, and 4 daily use: mean 0.43, s.d. 0.53),
and only one had previously used any eye-tracking-based user interfaces before (same
5-point scale: mean 0.14, s.d. 0.37).

## 7.2  User Study Results

The average SSQ score was $25.6 \pm 29.8$ (median 14.9), which is on par with other
VR applications that have been evaluated using SSQ (see, e.g., [27]). From TLX, we
used all categories (mental demand, physical demand, temporal demand, success, effort,
insecurity), on a 7-point scale where 0=Very Low and 6=Very High for the demand
metrics, and 0=Perfect, 6=Failure for the performance metrics. Users reported medium
scores for mental demand ($2.71 \pm 1.70$) and for effort ($2.86 \pm 1.68$), while reporting low
scores for physical demand ($1.86 \pm 1.95$), temporal demand ($1.57 \pm 0.98$), and insecurity
($1.14 \pm 1.68$). The participants judged themselves to have been rather successful with
the tracking tasks ($1.71 \pm 0.75$).

All questions asked related to software usability and acceptance are summarised in
Figure 4a. The users estimated that the Bionic Tracking method would yield a speedup
of a factor 2 to 10 ($3.33 \pm 6.25$) compared to tracking cells with a regular 2D interface,
and expressed high interest in using the method for their own tracking tasks ($3.43 \pm 0.53$;
5-point scale here and for the following: 0=No agreement, 4=Full agreement), as the
tracks created by it looked reasonable ($2.57 \pm 0.98$), it would provide an improvement
over their current methods ($3.14 \pm 0.90$), and they could create new cell tracks not only
with confidence ($2.86 \pm 0.69$), but also faster ($3.29 \pm 0.76$). Users found the software
relatively intuitive ($2.43 \pm 0.98$) and did not need a long time to learn how to use it
($0.59 \pm 0.79$), which they also remarked on the the follow-up interviews:

"It was so relaxing, actually, looking at this [cell] and just looking." (P2, the
user remarked further after the interview that the technique might prevent carpal
tunnel issues often encountered when tracking via mouse and keyboard.)

"I figured this could be like a super quick way to generate the [cell] tracks." (P7)

Furthermore, the user study showed that users tend to adjust playback speed more often than image size (in VR). After playing around with different settings – users could choose speeds from 1 to 20 volumes/second – all users interestingly settled on 4-5 volumes/second, corresponding to 200 to 250 ms of viewing time per timepoint, which coincides with the onset delay of smooth pursuit eye movements. Albeit having no or limited previous VR experience, the users did not feel irritated by the environment $(0.00 \pm 0.00)$ nor by the use of eye tracking $(0.29 \pm 0.49)$.

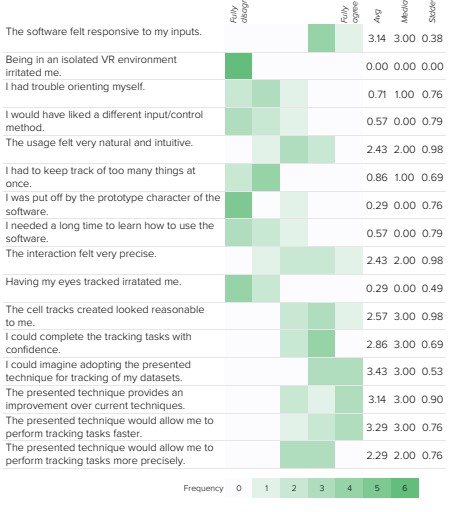

| | Avg | Median | Stddev |
|---|---|---|---|
| The software felt responsive to my inputs. | 3.14 | 3.00 | 0.38 |
| Being in an isolated VR environment irritated me. | 0.00 | 0.00 | 0.00 |
| I had trouble orienting myself. | 0.71 | 1.00 | 0.76 |
| I would have liked a different input/control method. | 0.57 | 0.00 | 0.79 |
| The usage felt very natural and intuitive. | 2.43 | 2.00 | 0.98 |
| I had to keep track of too many things at once. | 0.86 | 1.00 | 0.69 |
| I was put off by the prototype character of the software. | 0.29 | 0.00 | 0.76 |
| I needed a long time to learn how to use the software. | 0.57 | 0.00 | 0.79 |
| The interaction felt very precise. | 2.43 | 2.00 | 0.98 |
| Having my eyes tracked irratated me. | 0.29 | 0.00 | 0.49 |
| The cell tracks created looked reasonable to me. | 2.57 | 3.00 | 0.98 |
| I could complete the tracking tasks with confidence. | 2.86 | 3.00 | 0.69 |
| I could imagine adopting the presented technique for tracking of my datasets. | 3.43 | 3.00 | 0.53 |
| The presented technique provides an improvement over current techniques. | 3.14 | 3.00 | 0.90 |
| The presented technique would allow me to perform tracking tasks faster. | 3.29 | 3.00 | 0.76 |
| The presented technique would allow me to perform tracking tasks more precisely. | 2.29 | 2.00 | 0.76 |

Frequency 0 1 2 3 4 5 6

**(a)** Results of usability and acceptance-related question from the user study. Please note that the questions are formulated both positively and negatively.

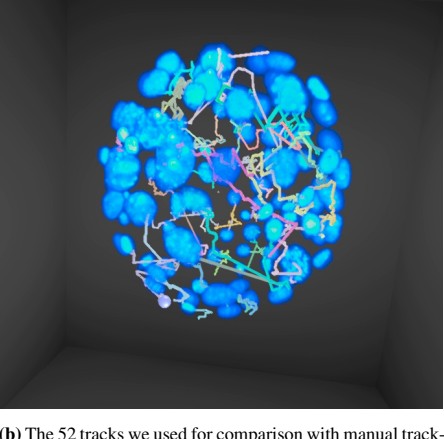

**(b)** The 52 tracks we used for comparison with manual tracking results visualised together with the volumetric data of one timepoint. This is the same view the user had, taken from within the VR headset. See the supplementary video for a demonstration of creating these tracks.

**Fig. 4:** User study and cell tracking results for the *Platynereis* dataset.

### 7.3   Comparison with Manual Tracking Results

To further characterize the performance of Bionic Tracking, we performed a comparison to manually annotated tracks. Our primary focus in this comparison is to assess the capacity of Bionic Tracking to recreate individual manually annotated tracks. We compared 52 tracks created by an expert annotator using Bionic Tracking (see Figure 4b) on the *Platynereis* dataset to their respective best matching ground truth tracks. We find that 25 of the 52 tracks have a distance score [32] that is less than 1 cell diameter, suggesting that these tracks will, on average, intersect the volume of their corresponding cell.

## 8   Discussion

We were able to show that gaze in VR can be used to reconstruct tracks of biological cells in 3D microscopy. Our method does not only accelerates the process, but makes

manual tracking tasks also easier and less demanding. Although our expert-based user study was rather small in size, limiting its statistical power, we believe that it provides an indication that the use of Bionic Tracking can improve the user experience and speed for cell tracking tasks, and that developing it further is worthwhile.

Even though the users had limited previous VR experience before, they were quickly able to create cell tracks with high confidence. Multiple users complimented the ergonomics of the technique, although it remains to be seen whether this would still be the case for longer (1h+) tracking sessions. With the projected speedups, however, it might not even be necessary to have such long sessions anymore (users indicated that for manual tracking, they would not do sessions longer than 3 to 4 hours, with the estimated speedups, this could be potentially reduced to just 20-90 minutes using Bionic Tracking).

For tracking large lineages comprising thousands of cells, Bionic Tracking on it own is going to be cumbersome, for combinatorial reasons. It can, however, augment existing techniques for parts of the tracking process, e.g., to track cells only in early stages of development, where they tend to have less well-defined shapes, or it may provide constraints and training data for machine-learning algorithms of automated tracking. Furthermore, Bionic Tracking could be used in conjunction with any automatic tracking algorithm that provides uncertainty scores in order to restrict gaze input to regions where the algorithm cannot perform below a given uncertainty threshold. This could be done, e.g., by superimposing a heatmap on the volume rendering to indicate to the user areas that need additional curation. Hybrid semi-automated/manual approaches are already among the most popular tools for challenging biological datasets [35].

## 9   Future Work and Limitations

In the future, we would like to integrate Bionic Tracking into an existing tracking software, such that it can be used by a general audience. Unfortunately, eye tracking-enabled HMDs are not yet widely available, but according to current announcements, this is likely to change. Current developments in eye tracking hardware and VR HMDs indicate falling prices in the near future, such that those devices might soon become more common, or even directly integrated into off-the-shelf HMDs. One could imagine just having one or two eye tracking-enabled HMDs as an institute, making them available to users in a bookable item-facility manner. At the moment, the calibration of the eye trackers can still be a bit problematic, but this is likely to improve in the future, too, with machine learning approaches making the process faster, more reliable, and more user-friendly.

In order for Bionic Tracking to become a tool that can be routinely used for research in biology, it will be necessary to implement interactions that allow the user to indicate certain events, like cell divisions. Such an interaction could for example include the user pressing a certain button whenever a cell division occurs, and then track until the next cell division. In such a way, the user can skip from cell division to cell division, literally applying divide-and-conquer for tracking (a part of) the cell lineage tree at hand. These additional features will enable the creation of entire cell lineage trees.

The design and evaluation of algorithms to detect and track entire lineage trees is currently an active focus in the systems biology community [32]. In this study, we have

used comparison algorithms from the Particle Tracking Challenge (PTC) [5], which were designed to compare single tracks. There are limitations when applying the PTC metric to compare cell tracking annotations. However, until additional tracking events—such as the aforementioned cell divisions—can be recorded with Bionic Tracking, PTC is the only metric that can be applied.

In our tests, we have still seen some spurious detections, which lead to tracks obviously not taken by the cell. This calls for more evaluations within crowded environments: While Bionic Tracking seems well suited for crowded scenes in principle – as users can, e.g., move around corners and are tracked by the HMD – it is not yet clear whether eye tracking is precise enough in such situations.

In addition, head tracking data from the HMD could be used to highlight the area of the volumetric dataset the user is looking toward (foveated rendering, [18, 4]), e.g., by dimming areas the user is not looking at. We have not yet explored foveation, but could imagine it might improve tracking accuracy and mental load.

## 10   Conclusion

We have presented *Bionic Tracking*, a new method for object tracking in volumetric image datasets, leveraging gaze data and virtual reality HMDs for biological cell tracking problems. Our method is able to augment the manual parts of cell tracking tasks in order to render them faster, more ergonomic, and more enjoyable for the user, while still generating high-quality tracks. Users estimated they could perform cell tracking tasks up to 10-fold faster with Bionic Tracking than with conventional, manual tracking methods.

As part of Bionic Tracking, we have introduced a method for graph-based temporal tracking, which enables to robustly connect gaze samples with cell or object detections in volumetric data over time.

The results from our research prototype have been very encouraging, and we plan to continue this line of research with further studies, extending the evaluation to more datasets and users, and adding an evaluation of the accuracy of the created cell tracks on datasets that have known associated ground truth. Furthermore, we would like to add Bionic Tracking to a pipeline where the gaze-determined cell tracks can be used to train machine-learning algorithms to improve automatic tracking results. Our prototype software is available as open-source software at *removed for double-blind review*.

## Acknowledgements

Removed for double-blind review.

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
