# OpenReview forum: "Bionic Tracking: Using Eye Tracking to Track Biological Cells in Virtual Reality"
_thecvf.com/ECCV/2020/Workshop/BIC — BIC 2020 Oral_

### Official Review · AnonReviewer2 · 2020-07-30
**Interesting direction for tracking of cells in time-lapse video based on eye-tracking, with a small but promising user study.**

**Rating:** 6
**Confidence:** 4

**Review:**

### Summary
- The paper proposes an approach for the “manual” tracking of 3d+t datasets of biological cells that is carried out in virtual reality and cells are tracked via eye. The approach, that the authors call Bionick Tracking, offers an alternative to the established method of manually tracking cells (and lineage trees) on a 2D screen with mouse click.

- The authors investigate the question whether using eye gaze and the movement in virtual reality can facilitate cell tracking. In order to be able to infer the cell track from the tracked eye gaze, they propose to use a graph-based algorithm. They carry out a study with seven users to test their set-up with regards to usability and accuracy.

- They find that all users overall had a positive tracking experience. The users also stated that they believe that tracking with Bionick Tracking speeds up the tracking process.

### Major strengths of the paper
- The paper is understandable with a clear line of thought; Limitations are clearly stated.
- The authors set-up a pipeline for eye-tracking of cells in time-lapse videos with a focus on using commodity hardware. With this focus, chances are higher that the setup will actually be adapted by other labs.
- The authors carried out a user study with promising results including very positive feedback from the users.
- The approach is novel and addresses important challenges in the cell tracking community (visualization of 3D time-lapse videos and annotation of cell tracks).
- The authors provide a video that nicely explains the usage of their setup and their algorithm, which facilitates the understanding of the entire paper.

### Major weaknesses of the paper
- The extraction of the path from the gaze involves smoothing, but how to handle real jumps or datasets that are difficult to register? In this context, the ground-truth path has to contain real jumps that would potentially be smoothed out by the algorithm that they propose in the paper.
- The study is limited as they only tested their setup on seven users, but this limitation is clearly stated.
- There is no quantitative comparison of annotation time of conventional methods versus their method. The observation that their method is faster in tracking cells than conventional methods is based on the user’s opinion. This is an important finding, but should be backed up with further quantitative experiments.

### Language
Some sentences use vague or colloquial language:
  - “The initial setting for the scale of the dataset is to make it appear about 2m big.” (about 2m big)
  - “and a kind of Midas touch problem [10] remains” (a kind)
  - “At the moment, the calibration of the eye trackers can still be a bit problematic” (a bit)
  - “One could imagine just having one or two eye tracking-enabled HMDs as an institute” (One could imagine just….)

Unclear sentences and minor mistakes
- Line 42: "The 3D images the lineage trees are usually created based on fluorescence microscopy images."
Unclear sentence;
- Line 94: Unclear sentence, please split into two sentences.
- Line 129: "The occur when following a stimilus" → They occur when following a stimulus
- Line 391: "This corresponds to distances larger than double the standard deviation of all distances the hedgehog."
Unclear sentence
- Line 539: "Our method does not only accelerates the process, but makes"; Incorrect grammar (--> Our method does not only accelerate the process …)




**Reviews Visibility:**

I agree that my anonymized review is made publicly visible, if the submission is accepted.

---

### Official Review · AnonReviewer1 · 2020-07-31
**A new approach for annotating 3D+T cell tracking data**

**Rating:** 6
**Confidence:** 4

**Review:**

Cell tracking is an important but challenging step for many biological research. By combining VR and eye tracking, the authors developed a new method for tracking cells in 3D and time. With the help of the two devices, users can generate cell trajectories by simply looking at a cell in a 3D movie. In the manuscript, detailed usage and user study data are provided. From what described, this new approach appears to be a great addition and can potentially make cell tracking annotation faster and enjoyable as stated by the authors.

Major comments:
- The manuscript is well written with details that are necessary, the limitations are clearly stated.
- Although it is hard to grasp the actual user experience without trying the device, the user study results seem promising and reflect a good overall experience.
- Although the authors provide user estimation of a 10x speed up with the new approach, it would be more convincing to actually measure the time for conventional methods and compare them with the new.
- Since the user can only look at one cell at a time with the new approach, this will likely limit the overall annotation throughput. Instead of focusing on generating trajectories cell-by-cell, I would encourage the authors to explore ways to use the new hardware to fix and curate trajectories generated by automated algorithms. Similar to what is mentioned in the end of the manuscript, it would be also interesting to see how this can be combined with machine-learning algorithms.

Minor comments:
- The provided supplementary video is quite helpful to the understanding of the approach, it would be helpful also to mention the video in the manuscript.
- When describing the hardware, could you provide detailed information about the eye tracking resolution? For users who are not familiar with the device, it is better to get some feeling about how accurate the eye tracking device is.
- Could you also discuss how the cell size impacts the tracking performance? When the cell is big, should the user look at the center of the cell? Is there an optimal display cell size for the device?


**Reviews Visibility:**

I agree that my anonymized review is made publicly visible, if the submission is accepted.

---

### Decision · Program_Chairs · 2020-07-31

Accept (Oral)